# The Effects of Different Sowing Density and Nitrogen Topdressing on Wheat Were Investigated under the Cultivation Mode of Hole Sowing

Yitao Sun [1,†], Wenlong Yang [1,†], Yilun Wu [1,†], Youhe Cui [1], Yongli Dong [1], Zhoujia Dong [2] and Jiangbo Hai [1,*]

1  College of Agronomy, Northwest A&F University, Yangling, Xianyang 712100, China; sunyitao@nwafu.edu.cn (Y.S.); yangwenlong2021@126.com (W.Y.); wuyilun21@163.com (Y.W.); cuiyouhe1999@163.com (Y.C.); 2008114433@nwafu.edu.cn (Y.D.)
2  Agricultural and Animal Husbandry Comprehensive Service Center, Tongren 811399, China; dzj678@163.com
*  Correspondence: haijiangbo@126.com; Tel.: +86-133-8922-1092
†  These authors contributed equally to this work.

**Abstract:** Hole sowing is a new and efficient cultivation method with few studies. This study investigated the effects of different sowing densities and nitrogen topdressing at the jointing stage on dry matter, quality, and yield under wheat hole sowing to provide a theoretical basis for integrating wheat fertilizer and density-supporting technology. In this study, a two-factor split-plot design was used. The sowing density was the main plot, and four levels were set: D1, D2, D3, and D4 (238, 327, 386, and 386 suitable seeds·m$^{-2}$). The four sowing levels were sown according to 8 grains/hole, 11 grains/hole, 13 grains/hole, and 16 grains/hole, respectively, with a row spacing of 25 cm and a hole spacing of 13.5 cm; the amount of nitrogen fertilizer applied at the jointing stage was the sub-area, with four levels: N1, N2, N3, and N4 (0, 60, 120, and 180 kg·ha$^{-1}$). After two years of experimental research, the following main conclusions are drawn: the use of high sowing density and nitrogen topdressing is helpful to improve the dry matter quality of wheat spikes at the maturing stage; the sowing density had significant or highly significant effects on protein content, starch content, and sedimentation value. The yield from 2018–2019 reached a maximum of 8448.67 kg·ha$^{-1}$ under D4N4 treatment, and the yield from 2019–2020 reached a maximum of 10,136.40 kg·ha$^{-1}$ under D4N3 treatment. Therefore, the combination of 225 kg·ha$^{-1}$ sowing density and 120–180 kg·ha$^{-1}$ nitrogen topdressing at the jointing stage can be used in field production, which can help improve wheat production potential. Similarly, understanding the interaction between wheat hole sowing and different sowing densities and nitrogen topdressing amounts provides a practical reference for high-yield wheat cultivation techniques.

**Keywords:** wheat; *Triticum aestivum* L.; hole sowing; cultivation techniques; yield

## 1. Introduction

With the growth of the population, food security has become a severe problem for the world. In 2015, among the world's 7.3 billion people, an estimated 654 million people were malnourished [1–3]. By 2019, 864 million people were considered malnourished. In order to meet global food demand, food production needs to increase by 70~100% by 2050 [4–6]. Wheat is an important food source for humans worldwide, with 20% of the world's wheat consumption by 50% of the world's poorest people [7–10]. More than 50% of the world's wheat comes from developing countries, and more land is planted for wheat than for any other crop in the world [11,12].

In wheat cultivation, sowing density and nitrogen fertilizer are critical factors affecting wheat population structure and yield formation [13–16]. Suitable sowing density can make wheat make full use of water, nutrients, and light energy [17,18], alleviate the competition

between populations and individuals, and help to construct a reasonable population structure [19,20]. Rational use of nitrogen fertilizer can promote the healthy growth of wheat, improve grain quality, increase yield, and achieve sustainable development of agriculture [15,21,22]. Many experts and scholars have carried out much research on the level of nitrogen supply in crops. If the application of chemical fertilizer is stopped, it will half the total global crop yield [23–25]. In addition, the unreasonable use of nitrogen fertilizer will also lead to environmental problems such as groundwater pollution [26], greenhouse effect, soil acidification [27], and so on. Therefore, the rational use of nitrogen fertilizer while achieving high yield and quality of wheat is significant for wheat production.

As a new cultivation technology, wheat hole sowing is an efficient agricultural technology integrating rain, drought resistance, and efficient utilization of light and heat resources [28,29]. Due to the characteristics of wheat hole sowing cultivation, each hole has a noticeable border effect. The outer wheat of each hole has more solar energy, better ventilation, and less nutrient competition than the inner wheat [30]. Therefore, in the actual field production, the boundary advantage of hole sowing itself helps to improve productivity and bring more economic benefits and value to people.

In this study, from 2018 to 2020, through wheat cultivation with the hole sowing method, its border effect was measured. Different amounts of nitrogen fertilizer were applied according to different sowing densities and jointing stages to explore the effects of different sowing densities and nitrogen topdressing amounts and their interaction on the dry matter, quality, and yield of wheat. We assumed that different sowing density, nitrogen topdressing, and their interaction would have different effects on dry matter of wheat spikes, grain quality, and yield. The objectives of this study were to: (1) explore the effects of different sowing density and nitrogen topdressing on dry matter of wheat spikes; (2) evaluate the effects of different sowing density and nitrogen topdressing on grain quality; (3) evaluate the effects of different sowing density and nitrogen topdressing on yield and components. This study's results will help provide new ideas and references for future research on wheat hole sowing to help scholars quickly lock in relevant knowledge and insights in the field.

## 2. Materials and Methods

### 2.1. Test Designs

This experiment was conducted at the Doukou Crop Experimental Demonstration Station of Northwest A & F University from 2018 to 2020. The experimental demonstration station is located in Xinglong Village, Yunyang Town, Jingyang County, Xianyang City, Shaanxi Province, China, 108°52′ E, 34°37′ N. The precipitation during the two-year growth period of wheat was 84.57 mm and 122.68 mm, and the average temperature was 9.53 °C and 10.65 °C, respectively (Figures 1 and 2). The soil in the test field was loam. Before sowing, 0–40 cm soil samples were randomly drilled at 5 points. After air drying, grinding, and screening, the soil's basic nutrient content was determined: organic matter content (potassium dichromate method) 18.02 g·kg$^{-1}$, total nitrogen content (inorganic and organic, semi-micromethod of Kay's fixed nitrogen) 1.39 g·kg$^{-1}$, available nitrogen content (nitric acid powder test method) 86.8 mg·kg$^{-1}$, available phosphorus content (ultraviolet spectrophotometry colorimetry) 16.83 mg·kg$^{-1}$, available potassium content (flare photometer) 232.07 mg·kg$^{-1}$, pH value 7.93, with medium fertility.

The 'XN805' wheat variety was selected as the experimental material. The variety is a semi-winter mid-early-maturity variety, with semi-stowing seedlings, dark green leaves, medium tillering ability, high panicle rate, medium winter cold resistance, medium late spring cold resistance, and average plant height of 66.9 cm and of a compact plant type. The main area was sowing density, and four sowing density levels were set: D1 (238 suitable seeds·m$^{-2}$), D2 (327 suitable seeds·m$^{-2}$), D3 (386 suitable seeds·m$^{-2}$), D4 (475 suitable seeds·m$^{-2}$). The sub-area was the amount of nitrogen topdressing at the jointing stage (P, K fixed), and four nitrogen fertilizer application levels were set. The nitrogen fertilizer (nitrogen content 46.4%) and the base fertilizer were wheat special slow-release fertilizer

(N: $P_2O_5$: $K_2O$ mass fraction 24: 15: 5) 750 kg·ha$^{-1}$, and the base fertilizer was applied once during rotary tillage. Nitrogen fertilizer without basal fertilizer was applied at the jointing stage: N1 (no urea), N2 (urea 60 kg·ha$^{-1}$), N3 (urea 120 kg·ha$^{-1}$), N4 (urea 180 kg·ha$^{-1}$). The sowing method was hole sowing. After calculating the four sowing density levels, the sowing was carried out according to 8 grains/hole, 11 grains/hole, 13 grains/hole, and 16 grains/hole, respectively. The row spacing was 25 cm and the hole spacing was 13.5 cm. Each plot was 3.5 m × 2 m = 7 m$^2$. Sowing was carried out manually on 5 October 2018 and 1 October 2019, weeding and pest control were carried out at different crop growth stages throughout the wheat growing season, and other management measures were taken to ensure consistency with local high-yielding farmland. During the experiment, the wheat was sown for 10 days, in mid-November, March, and May of the second year, and irrigated according to the actual situation in the field. The two-year processing was consistent and it was harvested on 4 June 2019 and 1 June 2020.

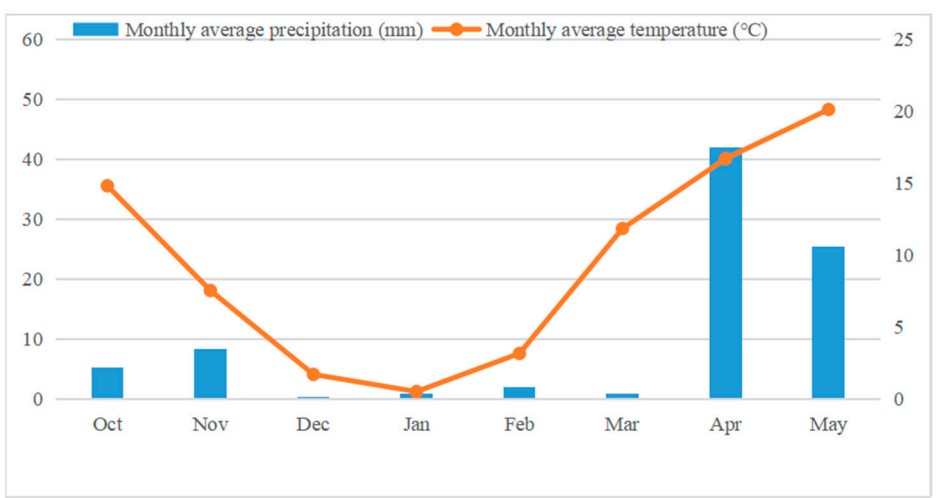

**Figure 1.** Total precipitation and monthly mean temperature during wheat growth stage from October 2018 to June 2019.

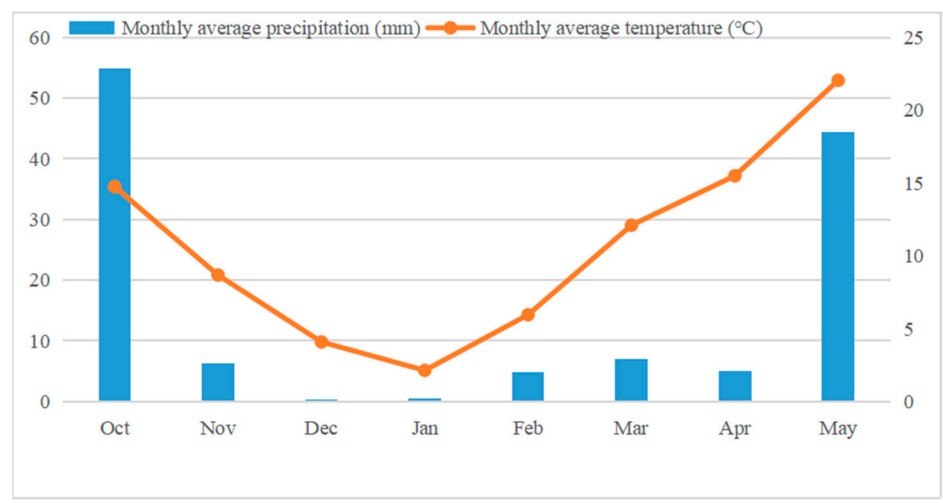

**Figure 2.** Total precipitation and monthly mean temperature during wheat growth stage from October 2019 to June 2020.

*2.2. Determination Items and Methods*

Dry matter of wheat spikes: 20 plants with uniform growth were randomly selected in each hole at the booting stage, heading stage, flowering stage, filling stage, and maturing stage. On the same day as harvesting, the wheat spikes were baked in an oven at 105 °C

for 30 min, then the temperature was reduced to 60–80 °C, and the drying was continued for about 8 h so that it was quickly dried and then removed. Finally, the sample continued to dry for 4 h, with weighing again until the weight was constant, then the final weight was measured.

Wheat grain quality: After two months of physiological after-ripening, the protein content, stability time, starch content, and sedimentation value of wheat grain samples after harvest were measured by the Danish FOSS Infratec TM 1241 (Manufactured by FOSS China Co., Ltd., Beijing, China) near-infrared grain quality analyzer.

Yield composition statistics: After the wheat matured, the effective panicles in the 1 m double-row sample section of each plot were counted. After harvest, the samples were sun-dried to remove impurities and a few plates of 1000 grains were weighed, which was repeated 3 times and the average value was taken for the 1000-grain weight. In each plot, 20 plants with uniform growth were randomly selected, and the grains per spike were counted to obtain the average value. Due to the small area of the plot, the hole sowing had an obvious border effect. In order to eliminate the influence of the border effect on the yield, 1 m$^2$ wheat was randomly taken from each plot in the middle and threshed with a thresher, dried in the sun, and weighed with an electronic balance to calculate the grain yield (kg·ha$^{-1}$).

### 2.3. Statistical Analysis of Data

Microsoft Office Excel 2021 and SPSS 26.0 were used for statistical analysis. RStudio was used for linear regression analysis, correlation analysis, and figure drawing. The significance level *(p < 0.05)* was used to judge the average difference by the minimum significant difference test.

## 3. Results

### 3.1. Effects of Different Sowing Density and Nitrogen Topdressing on Dry Matter of Wheat Spikes

The dry matter of wheat spikes at different stages (booting stage, heading stage, flowering stage, filling stage, and maturing stage) was measured and analyzed (Table 1). It can be seen from Table 1 that the effect of sowing density on the dry matter under different treatments designed in this experiment was very significant in the heading stage, flowering stage, filling stage, and maturing stage from 2018–2019 and 2019–2020. The effect of nitrogen topdressing amount on the dry matter under different treatments was highly significant at the filling stage and maturing stage from 2018–2019 and 2019–2020, and there were also significant differences between the heading stage and the flowering stage from 2019–2020. There was no significant difference in the dry matter of wheat spikes in different years; there were significant differences in heading stage, flowering stage, filling stage, and maturing stage of wheat between different years and sowing densities.

**Table 1.** Effects of different sowing density and nitrogen topdressing on dry matter of wheat spikes in different stages.

| Year | Sowing Density | Nitrogen Topdressing | Booting Stage | Heading Stage | Flowering Stage | Filling Stage | Maturing Stage |
|---|---|---|---|---|---|---|---|
| 2018–2019 | D1 | N1 | 1.12 ab | 1.35 cde | 1.65 def | 1.68 g | 1.75 k |
| | | N2 | 1.06 ab | 1.42 abcde | 1.78 bcdef | 1.89 fg | 2.08 jk |
| | | N3 | 1.13 ab | 1.56 abc | 1.79 bcdef | 2.21 bcd | 2.35 hij |
| | | N4 | 1.12 ab | 1.39 bcde | 1.74 cdef | 2.18 bcd | 2.42 ghij |
| | D2 | N1 | 0.98 ab | 1.23 de | 1.62 ef | 1.92 efg | 2.24 ij |
| | | N2 | 1.03 ab | 1.27 de | 1.58 f | 1.92 efg | 2.29 hij |
| | | N3 | 1.07 ab | 1.38 bcde | 1.74 cdef | 2.01 def | 2.59 defgh |
| | | N4 | 0.96 ab | 1.19 e | 1.93 abcd | 2.17 bcde | 2.53 efghi |
| | D3 | N1 | 1.12 ab | 1.48 abcd | 1.92 abcd | 2.07 cdef | 2.44 fghi |
| | | N2 | 0.94 b | 1.26 de | 1.99 abc | 1.99 def | 2.77 cdef |

**Table 1.** *Cont.*

| Year | Sowing Density | Nitrogen Topdressing | Booting Stage | Heading Stage | Flowering Stage | Filling Stage | Maturing Stage |
|---|---|---|---|---|---|---|---|
| | | N3 | 0.88 b | 1.42 abcde | 1.88 abcde | 2.13 cdef | 2.81 cde |
| | | N4 | 1.22 a | 1.36 bcde | 1.92 abcd | 2.39 b | 3.07 abc |
| | D4 | N1 | 0.93 b | 1.57 abc | 1.91 abcde | 2.19 bcd | 2.75 cdefg |
| | | N2 | 0.97 ab | 1.57 abc | 1.91 abcde | 2.3 bc | 2.92 bcd |
| | | N3 | 1.06 ab | 1.62 ab | 2.04 ab | 2.99 a | 3.19 ab |
| | | N4 | 1.22 a | 1.66 a | 2.13 a | 3.09 a | 3.39 a |
| | F value | FD | 1.117 | 14.17 *** | 10.608 *** | 54.44 *** | 214.04 *** |
| | | FN | 1.95 | 1.95 | 2.423 | 39.23 *** | 79.96 *** |
| | | FD × FN | 2.15 | 0.893 | 1.209 | 4.109 * | 0.047 |
| 2019–2020 | D1 | N1 | 0.98 cd | 1.17 f | 1.45 i | 1.83 f | 1.90 h |
| | | N2 | 1.06 bcd | 1.34 def | 1.66 ghi | 1.77 f | 2.04 gh |
| | | N3 | 1.13 bcd | 1.44 cd | 1.73 fghi | 1.89 ef | 2.14 gh |
| | | N4 | 1.16 abc | 1.42 cde | 1.8 defgh | 1.88 ef | 2.16 g |
| | D2 | N1 | 0.98 cd | 1.41 cde | 1.75 efghi | 1.96 ef | 2.07 gh |
| | | N2 | 1.03 bcd | 1.34 def | 1.59 hi | 1.83 f | 2.08 gh |
| | | N3 | 1.07 bcd | 1.41 cde | 1.62 hi | 1.86 f | 1.91 gh |
| | | N4 | 0.99 cd | 1.48 cd | 1.73 fghi | 2.09 de | 2.45 f |
| | D3 | N1 | 1.12 bcd | 1.54 bc | 1.96 cdefg | 2.24 cd | 2.52 ef |
| | | N2 | 0.94 cd | 1.24 ef | 1.81 defgh | 2.27 cd | 2.71 cde |
| | | N3 | 0.91 d | 1.31 def | 2.01 cdef | 2.43 c | 2.57 ef |
| | | N4 | 1.25 ab | 1.46 cd | 2.06 bcde | 2.41 c | 2.88 bcd |
| | D4 | N1 | 0.93 cd | 1.37 cdef | 2.13 bcd | 2.34 c | 2.64 def |
| | | N2 | 1 cd | 1.49 cd | 2.35 ab | 2.84 b | 2.95 bc |
| | | N3 | 1.09 bcd | 1.73 ab | 2.24 bc | 2.96 ab | 3.08 ab |
| | | N4 | 1.39 a | 1.81 a | 2.67 a | 3.08 a | 3.3 a |
| | F value | FD | 2.876 * | 8.932 *** | 46.67 *** | 91.733 *** | 105.9 *** |
| | | FN | 0.728 | 5.437 ** | 16.16 *** | 6.903 *** | 15.8 *** |
| | | FD × FN | 1.093 | 5.205 *** | 1.206 | 6.291 *** | 3.128 ** |
| | | FY | 1.058 | 1.428 | 1.878 | 2.234 | 2.467 |
| | | FY × FD | ns | ** | * | ** | * |
| | | FY × FN | ns | ns | ns | ns | ns |
| | | FY × FD × FN | ns | ns | ns | ns | ns |

Note: Y, D, and N represent different years, sowing density, and nitrogen topdressing, respectively. Different letters in the same column mean significant difference at 0.05. ns, not significant at 0.05 probability level; *, **, and *** refer to significant differences at 0.05, 0.01, and 0.001 level, same as Tables 2 and 3.

At the late filling stage, the assimilates of wheat plants are transported to the grains in large quantities, and their dry matter reaches the maximum at the maturity stage, eventually affecting their yield. Therefore, compared with other growth stages, the dry matter of the wheat maturity stage has a more significant impact. With the increase in sowing density, the overall performance of dry matter in the maturing stage was N4 > N3 > N2 > N1; with the increase in the amount of nitrogen, the overall performance of dry matter in the maturing stage was D4 > D3 > D2 > D1. As the dry matter of wheat spikes in the maturing stage was the most prominent, different sowing densities and nitrogen topdressing amounts were significantly different at this stage. In our study, we further explored the effects of sowing density (Figure 3) on the dry matter of wheat spikes at the maturing stage from 2018–2020 by linear regression analysis. Through the analysis of the two-year experiment, it can be found that the sowing density has a significant influence on the dry matter of wheat spikes, and it is significant.

The results showed that, at the maturity stage, the dry matter weight of wheat spikes treated with D4N4 was higher than that of other treatments, and the dry matter weight of wheat spikes treated with D1N1 was lower than that of other treatments.

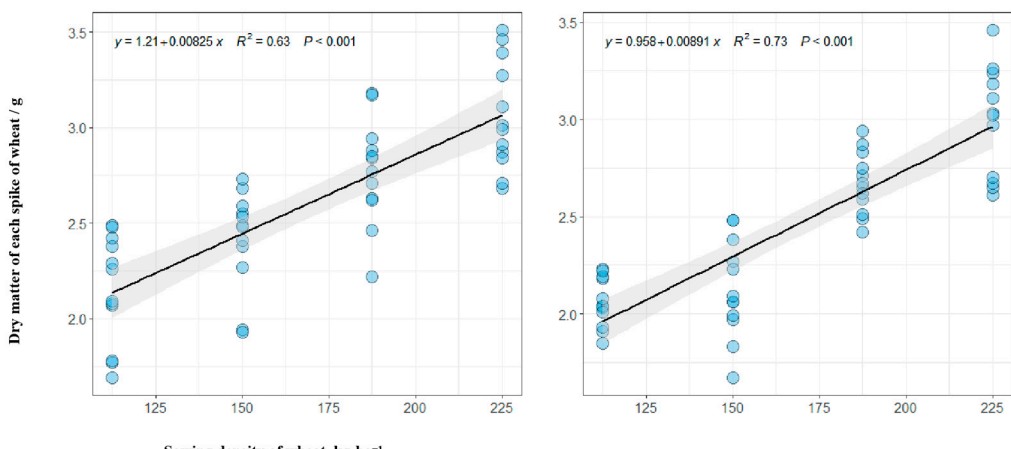

**Figure 3.** Linear regression analysis of different sowing density on dry matter of wheat spikes at the maturity stage from 2018–2020.

*3.2. Effects of Different Sowing Density and Nitrogen Topdressing on Grain Quality*

By analyzing the effects of different sowing densities and nitrogen topdressing on grain quality (Table 2), from 2018 to 2019, in terms of protein content, D2N4 treatment had the largest, 14.65%, and D4N3 treatment had the smallest, 13.46%; with the increase in sowing density, the protein content increased first and then decreased. The protein content was the highest at the D2 sowing density level, which was 14.42%, and the D2 level was significantly higher than the D3 level. The starch content increased with increased sowing density and nitrogen topdressing amount. Under the conditions of different sowing densities, the settlement value of grains increased first and then decreased with the increase in sowing density. Compared with D1, D3, and D4, the value of D2 increased by 3.50%, 8.93%, and 13.22% respectively.

From 2019 to 2020, the protein content increased first and then decreased with the increase in sowing density, and the specific performance was D3 > D2 > D1 > D4. Compared with D2, D1, and D4, the value of D3 increased by 0.20%, 6.73%, and 5.00%, respectively. The level of D3 was significantly higher than that of D4. The effect of nitrogen topdressing on protein content was D4 > D2 > D3 > D1, but there was no significant difference among different levels. The stabilization time increased first and then decreased with the increase in sowing density, and reached the maximum at the D2 level, and the stabilization time decreased with the increase in nitrogen topdressing. The starch content increased with the increase in sowing density and nitrogen topdressing amount. The sedimentation value of grains decreased with the increase in sowing density, and the value D1 was significantly higher than those of the other three sowing density levels, reaching 67.22 mL.

**Table 2.** Effects of different sowing density and nitrogen topdressing on grain quality of wheat at the maturity stage.

| Year | Sowing Density | Nitrogen Topdressing | Protein Content (%) | Stabilization Time (min) | Starch Content (%) | Settlement Value (mL) |
|---|---|---|---|---|---|---|
| 2018–2019 | D1 | N1 | 14.22 abc | 6.85 ab | 67.5 d | 50.77 abcd |
| | | N2 | 14.37 abc | 4.86 bcd | 68.47 abcd | 51.61 abc |
| | | N3 | 14.49 ab | 3.37 d | 67.68 cd | 52.06 abc |
| | | N4 | 14.53 ab | 2.97 d | 67.67 cd | 51.73 abc |
| | D2 | N1 | 14.56 ab | 8.10 a | 67.63 cd | 53.84 a |
| | | N2 | 14.38 abc | 4.31 bcd | 68.11 bcd | 50.66 abcd |
| | | N3 | 14.08 abc | 3.52 cd | 67.83 cd | 48.65 abcd |
| | | N4 | 14.65 a | 3.31 d | 67.41 d | 53.73 ab |

**Table 2.** *Cont.*

| Year | Sowing Density | Nitrogen Topdressing | Protein Content (%) | Stabilization Time (min) | Starch Content (%) | Settlement Value (mL) |
|------|------|------|------|------|------|------|
| | D3 | N1 | 13.76 abc | 8.26 a | 68.39 abcd | 47.25 abcd |
| | | N2 | 13.49 c | 3.17 d | 68.46 abcd | 43.60 d |
| | | N3 | 14.04 abc | 3.98 bcd | 68.76 abc | 49.51 abcd |
| | | N4 | 14.09 abc | 3.81 bcd | 68.48 abcd | 49.59 abcd |
| | D4 | N1 | 13.57 c | 7.25 abc | 68.77 abc | 46.25 bcd |
| | | N2 | 13.65 bc | 3.84 bcd | 69.38 a | 46.15 cd |
| | | N3 | 13.46 c | 2.25 d | 68.79 abc | 45.05 cd |
| | | N4 | 13.57 c | 2.89 d | 69.17 ab | 45.28 cd |
| | F value | FD | 9.731 * | 0.518 | 13.854 *** | 7.398 * |
| | | FN | 0.596 | 17.575 ** | 2.012 | 0.663 |
| | | FD × FN | 2.397 * | 3.945 ** | 0.508 | 2.090 * |
| 2019–2020 | D1 | N1 | 14.46 ab | 5.80 ab | 66.69 fg | 64.71 cd |
| | | N2 | 14.73 ab | 3.21 de | 67.38 defg | 67.35 ab |
| | | N3 | 15.41 ab | 3.23 de | 66.34 g | 67.85 ab |
| | | N4 | 15.58 a | 2.47 de | 67.57 def | 68.97 a |
| | D2 | N1 | 14.43 ab | 7.90 a | 66.97 efg | 63.63 cd |
| | | N2 | 15.56 ab | 4.47 bcd | 67.79 cde | 62.94 d |
| | | N3 | 15.41 ab | 4.63 bcd | 68.17 bcd | 64.22 cd |
| | | N4 | 15.47 ab | 3.04 de | 67.74 cdef | 65.88 bc |
| | D3 | N1 | 15.22 ab | 8.03 a | 68.72 abc | 63.22 cd |
| | | N2 | 15.49 ab | 4.00 cde | 67.79 cde | 64.07 cd |
| | | N3 | 15.25 ab | 3.45 de | 68.43 bcd | 63.48 cd |
| | | N4 | 15.02 ab | 3.04 de | 68.14 bcd | 63.75 cd |
| | D4 | N1 | 14.49 ab | 6.59 abc | 69.1 ab | 62.61 d |
| | | N2 | 14.67 ab | 3.95 cde | 69.05 ab | 63.18 cd |
| | | N3 | 14.28 b | 1.52 e | 69.13 ab | 63.48 cd |
| | | N4 | 14.66 ab | 2.47 de | 69.5 a | 63.41 cd |
| | F value | FD | 3.193 * | 2.107 | 28.464 *** | 19.318 ** |
| | | FN | 1.68 | 23.404 ** | 0.761 | 3.835 * |
| | | FD × FN | 1.552 | 5.425 ** | 2.524 * | 5.343 ** |
| | | FY | 14.533 *** | 4.392 | 68.157 | 56.828 *** |
| | | FY × FD | ns | ns | ns | ns |
| | | FY × FN | ns | *** | ns | ns |
| | | FY × FD × FN | ns | ns | ns | ns |

The results showed that sowing density had significant effects on protein content, starch content, and settlement value but did not significantly affect stabilization time from 2018 to 2020. The amount of nitrogen topdressing had a significant effect on stabilization time. The interaction between sowing density and nitrogen topdressing significantly affected protein content, stabilization time, and settlement value from 2018–2019. From 2019–2020, it significantly impacted stabilization time, starch content, and settlement value. Different years had significant differences in protein content and settlement value, and the interaction between different years and nitrogen topdressing showed significant differences in stabilization time.

### 3.3. Effects of Different Sowing Density and Nitrogen Topdressing on Yield and Components

By analyzing the effects of different sowing densities and nitrogen topdressing rates on yield (Figures 4 and 5) and yield components (Table 3), it was observed that sowing density had a significant effect on grain per spike, effective spikes, and yield from 2018–2020. The amount of nitrogen topdressing only had a significant effect on grain per spike and yield from 2018–2019. The interaction between sowing density and nitrogen application rate had significant effects on grain per spike, effective spikes, and yield.

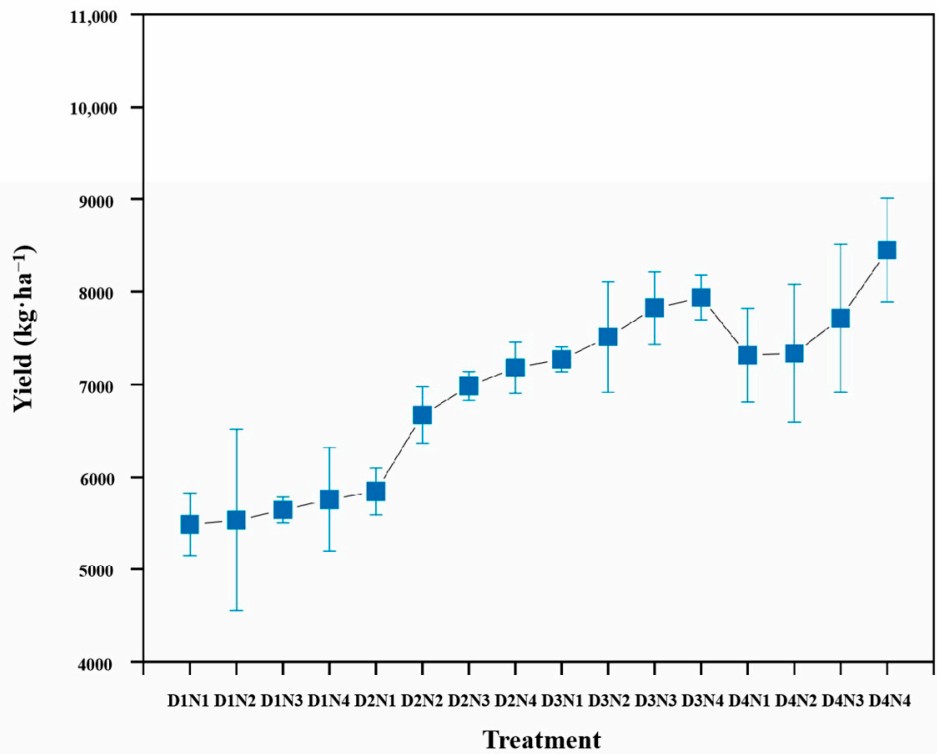

**Figure 4.** The dynamic changes in wheat yield under different sowing density and nitrogen topdressing from 2018–2019.

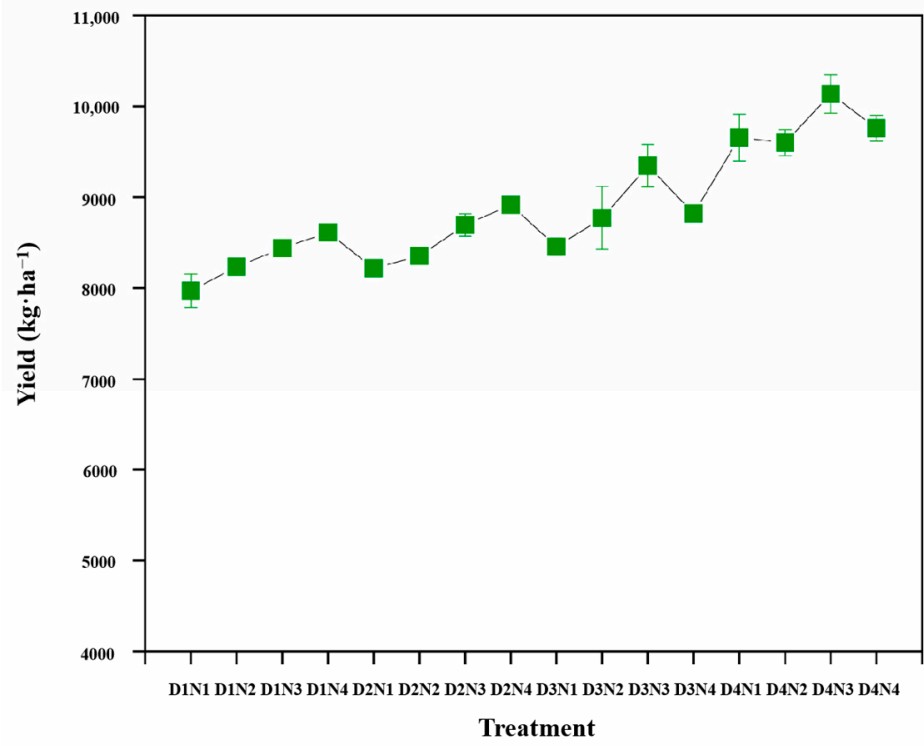

**Figure 5.** The dynamic changes in wheat yield under different sowing density and nitrogen topdressing from 2019–2020.

**Table 3.** Effects of different sowing density and nitrogen topdressing on yield and yield components.

| Year | Sowing Density | Nitrogen Topdressing | Grain per Spike | Thousand-Grain Weight (g) | Effective Spikes ($\times 10^4 \cdot ha^{-1}$) | Yield (kg·ha$^{-1}$) |
|---|---|---|---|---|---|---|
| 2018–2019 | D1 | N1 | 37.30 bc | 47.05 | 462.52 d | 5491.63 e |
| | | N2 | 38.46 ab | 46.57 | 462.01 d | 5536.10 de |
| | | N3 | 38.78 a | 46.21 | 471.09 d | 5647.27 de |
| | | N4 | 38.82 a | 46.51 | 476.74 d | 5758.43 de |
| | D2 | N1 | 37.18 bc | 46 | 516.04 cd | 5847.37 de |
| | | N2 | 37.37 bc | 45.98 | 515.59 cd | 6670.00 cd |
| | | N3 | 37.66 abc | 46.44 | 518.48 cd | 6981.27 bc |
| | | N4 | 37.84 abc | 46.47 | 521.59 bc | 7181.379 bc |
| | D3 | N1 | 36.63 cd | 47.74 | 578.96 bc | 7270.30 abc |
| | | N2 | 36.87 cd | 45.99 | 580.29 bc | 7514.872 abc |
| | | N3 | 37.18 bc | 45.62 | 594.01 ab | 7826.17 abc |
| | | N4 | 37.26 bc | 46.27 | 604.87 ab | 7937.40 ab |
| | D4 | N1 | 35.81 d | 46.4 | 624.76 ab | 7314.60 abc |
| | | N2 | 35.78 d | 45.02 | 657.93 a | 7336.83 abc |
| | | N3 | 35.80 d | 45.33 | 634.60 ab | 7714.97 abc |
| | | N4 | 36.85 cd | 45.81 | 643.18 ab | 8448.67 a |
| | F value | FD | 22.766 ** | 1.887 | 51.404 ** | 30.285 ** |
| | | FN | 4.104 * | 1.357 | 1.298 | 4.165 * |
| | | FD × FN | 5.749 ** | 1.554 | 14.061 ** | 7.199 ** |
| 2019–2020 | D1 | N1 | 33.17 ab | 46.64 ab | 526.06 d | 7972.82 f |
| | | N2 | 33.76 a | 46.13 ab | 553.73 cde | 8235.37 ef |
| | | N3 | 33.93 a | 48.13 a | 536.27 cd | 8441.39 cdef |
| | | N4 | 33.56 a | 47.51 ab | 551.61 cd | 8614.47 bcdef |
| | D2 | N1 | 32.13 abc | 47.82 ab | 562.62 bcde | 8220.78 ef |
| | | N2 | 32.26 abc | 47.12 ab | 558.95 bcde | 8356.01 def |
| | | N3 | 31.57 abcd | 46.92 ab | 565.78 abcde | 8695.35 bcdef |
| | | N4 | 31.52 abcd | 46.88 ab | 574.29 abcde | 8916.96 abcdef |
| | D3 | N1 | 30.29 abcd | 46.10 ab | 578.07 abcde | 8457.04 cdef |
| | | N2 | 30.88 abcd | 45.51 b | 591.50 abcde | 8772.18 bcdef |
| | | N3 | 30.50 abcd | 45.27 b | 595.80 abcd | 9346.59 abcde |
| | | N4 | 29.43 bcd | 45.50 b | 603.10 abcd | 8821.08 bcdef |
| | D4 | N1 | 28.32 d | 46.85 ab | 631.22 a | 9656.50 abc |
| | | N2 | 29.20 cd | 46.24 ab | 621.45 abc | 9599.63 abcd |
| | | N3 | 32.64 abc | 46.351 ab | 626.31 ab | 10136.40 a |
| | | N4 | 31.95 abcd | 46.69 ab | 622.16 abc | 9758.71 ab |
| | F value | FD | 7.556 ** | 5.796 ** | 7.122 ** | 12.187 ** |
| | | FN | 0.822 | 0.759 | 0.172 | 1.943 |
| | | FD FN | 2.362 ** | 2.230 * | 2.267 * | 2.987 ** |
| | | FY | 34.397 *** | 46.409 | 567.549 | 7889.954 *** |
| | | FY × FD | ns | ns | ns | ns |
| | | FY × FN | ns | ns | ns | ns |
| | | FY × FD × FN | ns | ns | ns | ns |

From 2018 to 2019, with the increase in sowing density, the number of grains per spike and 1000-grain weight decreased gradually, and the number of effective spikes increased continuously. In terms of the number of grains per spike, the high sowing density (D4) significantly decreased it by 6.32% compared with the low sowing density (D1). There was no significant difference in 1000-grain weight among different levels. The number of effective spikes in the D4 treatment increased significantly by 6.65%, 23.17%, and 39.13%, respectively, compared with D3, D2, and D1. With the increase in nitrogen topdressing, the number of grains per spike increased gradually, and the specific performance of 1000-grain weight was N1 > N4 > N2 > N3. The number of effective spikes increased first and then decreased, and the number of effective spikes under N2 treatment was the highest, which was 525.86 kg·ha$^{-1}$. From 2019 to 2020, with the increase in sowing density, the number of grains per spike decreased gradually, and the level of D1 was significantly higher than that

of other levels. The 1000-grain weight performance was D2 > D1 > D4 > D3. The effective spike number of the D4 treatment was significantly higher than that of D1 by 17.70%, and there was no significant difference among the other three sowing density levels.

From 2018 to 2019, the yield increased with the increase in sowing density. Compared with D3, D2, and D1, the value of D4 treatment increased by 66.59 kg·ha$^{-1}$, 1033.77 kg·ha$^{-1}$, and 2095.41 kg·ha$^{-1}$, respectively. The yield increased with the increase in nitrogen application. Compared with N3, N2, and N1, the value of N4 treatment increased by 289.05 kg·ha$^{-1}$, 567.02 kg·ha$^{-1}$, and 850.49 kg·ha$^{-1}$, respectively. From 2019 to 2020, there was a positive correlation between sowing density and yield. Compared with D1, D2, and D3, the value of D4 increased by 17.70%, 14.53%, and 10.51%, respectively. There was no significant difference among D1, D2, and D3 levels. The yield increased first and then decreased with the increase in nitrogen application. D4N3 treatment reached the maximum value of 10136.40 kg·ha$^{-1}$.

*3.4. Correlation Analysis of Different Indexes of Wheat*

Correlation analysis of different wheat indicators from 2018 to 2020 was carried out (Figures 6 and 7). It can be seen from Figure 6 that the sowing density was significantly positively correlated with effective spikes, starch content, and yield from 2018 to 2019. Sowing density was significantly negatively correlated with grain per spike, protein content, and settlement value. Nitrogen topdressing was significantly positively correlated with grain dry matter. It was significantly negatively correlated with stabilization time. From 2019 to 2020, sowing density was significantly positively correlated with effective spikes, starch content, yield, and grain dry matter. The sowing density was significantly negatively correlated with settlement value and grain per spike. Nitrogen topdressing was only significantly negatively correlated with stabilization time.

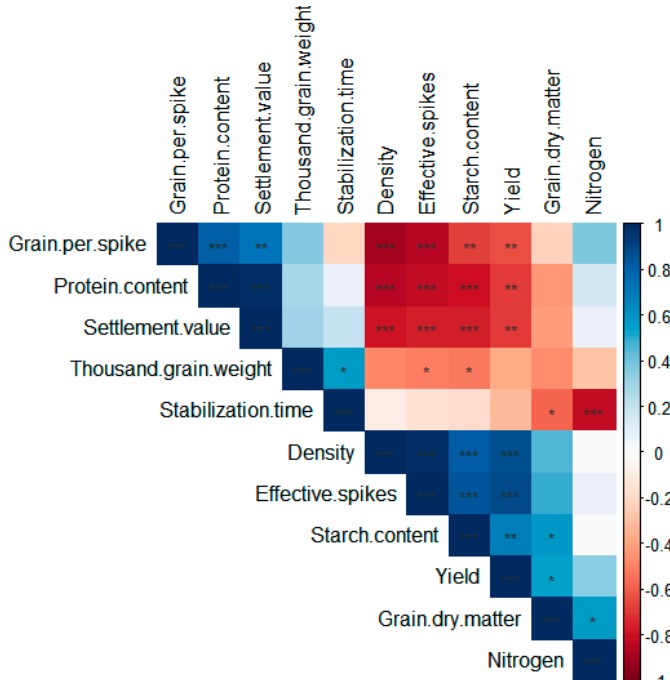

**Figure 6.** Correlation analysis of different wheat indexes from 2018–2019 (different colors in the figure represent positive and negative correlation, and color depth represents the correlation size. The bluer the color, the greater the positive correlation coefficient; the redder the color, the greater the negative correlation coefficient. X axis and Y axis represent different indexes, r values in the figure are in different colors, *: $p \leq 0.05$, **: $p \leq 0.01$, ***: $p \leq 0.001$, the same as Figure 7).

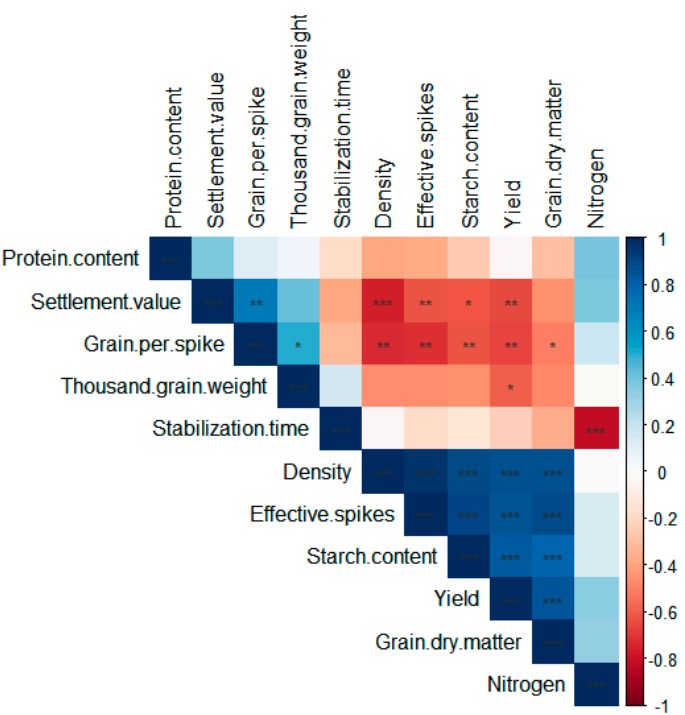

**Figure 7.** Correlation analysis of different wheat indexes from 2019–2020.

In summary, the increase in sowing density mainly promoted effective spikes, starch content, and yield and inhibited settlement value and grain per spike. The increase in nitrogen topdressing amount mainly inhibited the stabilization time.

## 4. Discussion

Sowing density is a limiting factor for plants to obtain environmental resources [31]. It is considered to be one of the most influential cultivation methods for grain yield and other agronomic traits. Changes in sowing density are particularly important in wheat crops and have a direct impact on grain yield and its components [14]. The dynamics of nitrogen and its loss trend create a challenging environment for the effective management of this nutrient in topdressing [32], which is mainly due to various reactions and instability in the soil. The low efficiency of nitrogen is attributed to the volatilization, leaching, and surface runoff of ammonia [20]. Some studies have found the effects of sowing density and nitrogen on crops. For example, Kanwal et al. [33], by evaluating the effects of different sowing densities and nitrogen doses on oat forage yield, found that the interaction of sowing density and nitrogen amount significantly changed the yield and quality attributes of oat green forage. The sowing rate of forage oat crops should be 90 kg·ha$^{-1}$ and supplemented with 120 kg·ha$^{-1}$ nitrogen, producing a higher yield, better quality, and better return.

Our research group has previously proved that hole sowing has an excellent effect on the growth characteristics of wheat by comparing the wheat hole sowing method with the traditional sowing method. Wu et al. [28] studied the effects of different sowing methods (drill sowing, wide sowing, and hole sowing) on the yield and quality of wheat. It was found that the hole sowing treatment increased the flag leaf area of wheat, the nitrogen application increased the dry matter quality of the above-ground part of the hole sowing treatment, and the actual yield of the hole sowing treatment was the highest. However, most of the field experiments on wheat sowing density and nitrogen topdressing in the past were carried out by drilling technology and the influence of the hole sowing cultivation method was not explored [34–37]. Under the conditions of this experiment, the density had a very significant effect on the number of effective spikes and yield. Increasing the sowing density would reduce the number of grains per spike and thousand grain weight, significantly increase the number of effective spikes per unit area, and expand the number

of populations, which could compensate for the lack of individuals. Increasing the amount of topdressing nitrogen had little effect on protein content and settlement value, which may be due to the high nutrient content in the soil before sowing in this experiment, so topdressing had little effect on the experiment. The results of our experiment were also slightly different between years. Different years had significant effects on dry matter of wheat spikes, protein content, settlement value, grain per spike, and yield. The dry matter of wheat spikes, the number of effective spikes per unit area, yield, protein content, and settlement value of each treatment from 2018–2019 were lower than those from 2019–2020. The main reason may be that, from 2019–2020, the precipitation and average temperature during the wheat growth period were higher than from 2018–2019, and abundant rainfall and suitable temperature were conducive to crop growth and development.

## 5. Conclusions

After two years of research on the use of different sowing densities and nitrogen topdressing amounts of wheat under hole sowing conditions, we found that field production can use a combination of a sowing density of 475 suitable seeds·m$^{-2}$ and 120–180 kg·ha$^{-1}$ of nitrogen topdressing at the jointing stage, which can fully tap the production potential of wheat. The experimental results fill the gap in wheat research on the cultivation method of hole sowing and provide valuable references and help for future researchers. In addition, there are still some limitations and deficiencies in this experimental study. The experiment was only over a two-year research period, and due to the significant difference in climatic conditions between the two years, although the overall trend is consistent, the regularity and universality of individual index changes are not strong. It is necessary to further carry out long-term positioning experiments to more accurately grasp and lay a theoretical basis and technical support for fully tapping wheat's high-quality and high-yield potential under hole sowing conditions.

**Author Contributions:** Y.S.: Investigation, Methodology, Writing—original draft, W.Y.: Writing—original draft, Y.W.: Methodology, Y.C.: Writing—review and editing, Y.D.: Writing—review and editing, Z.D.: Methodology, J.H.: Project administration, Funding acquisition. All authors have read and agreed to the published version of the manuscript.

**Funding:** The research was financially supported by the Ecological Security and Bioremediation Mechanism of Saline-alkali Soil Improvement in the Middle Yellow River (No. DL2021172002L).

**Institutional Review Board Statement:** Not applicable.

**Data Availability Statement:** Not applicable.

**Acknowledgments:** The authors express their special gratitude to the funding source for the financial assistance and are also thankful to the anonymous reviewers for their constructive and valuable comments on earlier versions of this research article.

**Conflicts of Interest:** The authors declare no conflict of interest.

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
