# Peer review of "The Effects of Different Sowing Density and Nitrogen Topdressing on Wheat Were Investigated under the Cultivation Mode of Hole Sowing"

_agronomy, doi:10.3390/agronomy13071733_

Round 1
Reviewer 1 Report
I am concerned about the acceptance of this paper. Indeed, most of the previous field experiments on wheat seeding density and N top dressing have been conducted in drill sowing, and the effect of hole sowing cultivation methods has not been examined. The present experiment is on the relationship between seeding density and N-top dressing. Many readers are interested in the characteristics of seeding density and N-top dressing in hole sowing cultivation. Comparing hole sowing with drill sowing is useful in examining sowing methods. However, in this experiment, only hole seeding was used as the seeding method. The changes in stems after fertilization are reasonable, but not new information for conventional drill sowing. The objectives and results do not seem appropriate. Multiple control of fertilizer and seeding density is a major effort in crop management and this experiment is an important study. The results and discussion are not questionable. Therefore, we responded that it is acceptable.Author Response
Response to comments
Dear Reviewer
We are grateful to the Journal of Agronomy for providing us with the valuable opportunity to revise and modify our manuscript. We would like to thank the reviewers for their careful review and comments, and for your hard work. All the comments have been carefully revised and highlighted in red color for your convenience. We shall look forward to hearing from you and hope that you will consider our work again and give us the opportunity to communicate further with the reviewers as needed.
In conclusion, we have tried our best to improve the manuscript and have made comprehensive changes in this paper. After reading the latest version of our manuscript, we hope you will have a better understanding of this manuscript. We earnestly appreciate the reviewers’ work and we hope that the changes and corrections will meet with your approval. Once again, thank you very much for your comments and suggestions.
Yours sincerely,
The authors
Reviewer 2 Report
The authors presented a paper with an interesting topic, especially in terms of the seeding method they chose to investigate.
However, the text needs extensive proofreading by a native English speaker and should more closely follow the form of writing scientific publications in terms of grammar, syntax, tense and person.
The introduction needs a better review of the literature and a clearer statement of the purpose of the paper.
In methods and materials, it would be useful to present monthly diagrams for the distribution of precipitation and temperature variation. It is also necessary to refer to the methods of analysis of the soil parameters and a better presentation of the data (e.g. the total nitrogen is only inorganic or organic as well)
There are some serious issues in the results that should be fixed.
First, the statistical analysis has been done without calculating and analyzing the interaction of year x D, year x N and year x N x D. The effect of the year factor appears to be significant, especially in yield performance, and these interactions should be analyzed for all measured and calculated parameters as well as their correlations (one corellation matrix for both years).
Secondly, the measurement of specific weight is absent from the quality characteristics. It is one of the most important features for determining wheat grain market value and all Foss machines routinely measure it. If authors have availible data they should present them.
Thirdly, authors did not have any calculation for the utilization of the supplied nitrogen by the crop (eg N harvest index, N use efficiency). This lack, combined with the very large amount of N as basal fertilization, does not allow reliable conclusions to be drawn as to the most correct choice of combination of sowing density and dose of N surface fertilization. The goals of the paper include the correct use of N to reduce the environmental impact, but the authors arbitrarily suggest the highest values, in a "the more, the better" logic.
The discussion should be rewritten. The authors in the first two paragraphs simply review the literature on other grains without linking or commenting on their own results. In the third paragraph, they comment on their own results without linking them to any literature references or previous corresponding work.
The conclusions should also be rewritten after the most careful statistical analysis and the most careful N utilization analysis and avoiding generalizations and boastful statements like "Our research helps to ensure food security".
Text needs extensive proofreading by a native English speaker and should more closely follow the form of writing scientific publications in terms of grammar, syntax, tense and person.
Author Response
Response to comments
Dear Reviewer
We are grateful to the Journal of Agronomy for providing us with the valuable opportunity to revise and modify our manuscript. We would like to thank the reviewers for their careful review and comments, and for your hard work. All the comments have been carefully revised and highlighted in red color for your convenience. We shall look forward to hearing from you and hope that you will consider our work again and give us the opportunity to communicate further with the reviewers as needed.
Response to reviewer comments:
- However, the text needs extensive proofreading by a native English speaker and should more closely follow the form of writing scientific publications in terms of grammar, syntax, tense and person.
Response: Thanks for pointing out the flaw in the manuscript, We have modified and improved the language and grammar of the article.
- The introduction needs a better review of the literature and a clearer statement of the purpose of the paper.
Response: Thanks for pointing out the flaw in the manuscript,We have adjusted and modified the introduction.“In wheat cultivation, sowing density and nitrogen fertilizer are critical factors affecting wheat population structure and yield formation [13-16]. Suitable sowing density can make wheat make full use of water, nutrients, and light energy [17; 18], alleviate the competition between populations and individuals, and help to construct a reasonable population structure [19; 20]. Rational use of nitrogen fertilizer can promote the healthy growth of wheat, improve grain quality, increase yield, and achieve sustainable development of agriculture [15; 21; 22]. Many experts and scholars have done much research on the level of nitrogen supply in crops. If the application of chemical fertilizer is stopped, it will lead to half of the total global crop yield [23-25]. In addition, the unreasonable use of nitrogen fertilizer will also lead to environmental problems such as groundwater pollution [26], greenhouse effect, soil acidification [27], and so on. Therefore, the rational use of nitrogen fertilizer while achieving high yield and quality of wheat is significant for wheat production.”(Lines 39-51)
- In methods and materials, it would be useful to present monthly diagrams for the distribution of precipitation and temperature variation. It is also necessary to refer to the methods of analysis of the soil parameters and a better presentation of the data (e.g. the total nitrogen is only inorganic or organic as well)
Response: Thanks for pointing out the flaw in the manuscript, We add Figure 1 and Figure 2.
Figure 1. Total precipitation and monthly mean temperature during wheat growth stage from October 2018 to June 2019
Figure 2. Total precipitation and monthly mean temperature during wheat growth stage from October 2019 to June 2020
- First, the statistical analysis has been done without calculating and analyzing the interaction of year x D, year x N and year x N x D. The effect of the year factor appears to be significant, especially in yield performance, and these interactions should be analyzed for all measured and calculated parameters as well as their correlations (one corellation matrix for both years).
Response: Thanks for pointing out the flaw in the manuscript, since the precipitation and average temperature of wheat growth period in 2019-2020 are higher than those in 2018-2019, abundant rainfall and suitable temperature are conducive to crop growth and development. Therefore, we mainly consider the impact of the overall trend of 2 years on wheat. The year of the test is only 2 years, and it is impossible to predict how the year will affect the test.
- Secondly, the measurement of specific weight is absent from the quality characteristics. It is one of the most important features for determining wheat grain market value and all Foss machines routinely measure it. If authors have availible data they should present them.
Response: We mainly consider the yield and hole sowing, so we do not measure the specific weight.
- Thirdly, authors did not have any calculation for the utilization of the supplied nitrogen by the crop (eg N harvest index, N use efficiency). This lack, combined with the very large amount of N as basal fertilization, does not allow reliable conclusions to be drawn as to the most correct choice of combination of sowing density and dose of N surface fertilization. The goals of the paper include the correct use of N to reduce the environmental impact, but the authors arbitrarily suggest the highest values, in a "the more, the better" logic.
Response: Thanks for pointing out the flaw in the manuscript, We have modified and supplemented this part. “As a new cultivation technology, wheat hole sowing is an efficient agricultural technology integrating rain, drought resistance, and efficient utilization of light and heat resources [36; 37]. Due to the characteristics of wheat hole sowing cultivation, each hole of wheat has a noticeable border effect. The outer wheat of each hole has more solar energy, better ventilation, and less nutrient competition than the inner wheat [28]. Therefore, in the actual field production, the boundary advantage of hole sowing itself helps to improve productivity and bring more economic benefits and value to people.
In this study, from 2018 to 2020, through wheat cultivation in the hole sowing method, its border effect is exerted. Different amounts of nitrogen fertilizer are applied according to different sowing densities and jointing stages to explore the effects of different sowing densities and nitrogen topdressing amounts and their interaction on the dry matter, quality, and yield of wheat. This study's results will help provide new ideas and references for future research on wheat hole sowing to help scholars quickly lock in relevant knowledge and insights in the field.”(Lines 52-65)
- The discussion should be rewritten. The authors in the first two paragraphs simply review the literature on other grains without linking or commenting on their own results. In the third paragraph, they comment on their own results without linking them to any literature references or previous corresponding work.
Response: Thanks for pointing out the flaw in the manuscript, We have modified and supplemented this part.“Sowing density is a limiting factor for plants to obtain environmental resources [29]. It is considered to be one of the most influential cultivation methods for grain yield and other agronomic traits. Changes in sowing density are particularly important in wheat crops, which have a direct impact on grain yield and its components [14]. The dynamics of nitrogen and its loss trend create a challenging environment for the effective management of this nutrient in topdressing [30], which is mainly due to various reactions and instability in the soil. The low efficiency of nitrogen is attributed to the volatilization, leaching, and surface runoff of ammonia [20]. Some studies have found the effects of sowing density and nitrogen on crops. For example Kanwal, A et al. [31], by evaluating the effects of different sowing densities and nitrogen doses on oat forage yield, it was found that the interaction of sowing density and nitrogen amount significantly changed the yield and quality attributes of oat green forage. The sowing rate of forage oat crops should be 90 kg·ha −1 and supplemented with 120 kg·ha −1 nitrogen. Higher yield, better quality, and better return.
Our research group has previously proved that hole sowing has an excellent effect on the growth characteristics of wheat by comparing the wheat hole sowing with the traditional sowing method. Wu et al. [36] studied the effects of different sowing methods (drill sowing, wide sowing, and hole sowing) on the yield and quality of wheat. It was found that the hole sowing treatment increased the flag leaf area of wheat, and the nitrogen application increased the dry matter quality of the above-ground part of the hole sowing treatment, and the actual yield of the hole sowing treatment was the highest. However, most of the field experiments on wheat sowing density and nitrogen topdressing in the past were carried out by drilling technology, and the influence of the hole sowing cultivation method was not explored [32-35]. Under the conditions of this experiment, the density had a very significant effect on the number of effective spikes and yield. Increasing the sowing density would reduce the number of grains per spike and thousand grain weight, significantly increase the number of effective spikes per unit area, and expand the number of populations, which could compensate for the lack of individuals. Increasing the amount of topdressing nitrogen had little effect on protein content and settlement value, which may be due to the high nutrient content in the soil before sowing in this experiment, so topdressing had little effect on experiment. The results of our experiment were also slightly different between years. The dry matter of wheat spike, the number of effective spikes per unit area, yield, protein content, and settlement value of each treatment in 2018-2019 were lower than those in 2019-2020. The main reason may be that in 2019-2020, the precipitation and average temperature during the wheat growth period were higher than in 2018-2019, and abundant rainfall and suitable temperature were conducive to crop growth and development. ”(Lines 285-321)
- The conclusions should also be rewritten after the most careful statistical analysis and the most careful N utilization analysis and avoiding generalizations and boastful statements like "Our research helps to ensure food security".
Response: Thanks for pointing out the flaw in the manuscript, We modified the conclusion.“After two years of research on the use of different sowing densities and nitrogen topdressing amounts of wheat under hole sowing conditions, we found that field production can use a nitrogen density combination of 475 suitable seeds·m −2 sowing density and 120 - 180 kg·ha -1 nitrogen topdressing at the jointing stage, which can fully tap the production potential of wheat. The experimental results fill the gap in wheat research under the cultivation method of hole sowing and provide valuable references and help for future researchers. In addition, there are still some limitations and deficiencies in this experimental study. The experiment is only a two-year research period, and due to the significant difference in climatic conditions between the two years, although the overall trend is consistent, the regularity and universality of individual index changes are not strong. It is necessary to further carry out long-term positioning experiments to more accurately grasp and lay a theoretical basis and technical support for fully tapping wheat's high-quality and high-yield potential under hole sowing conditions. ”(Lines 323-337)
In conclusion, we have tried our best to improve the manuscript and have made comprehensive changes in this paper. After reading the latest version of our manuscript, we hope you will have a better understanding of this manuscript. We earnestly appreciate the reviewers’ work and we hope that the changes and corrections will meet with your approval. Once again, thank you very much for your comments and suggestions.
Yours sincerely,
The authors
Reviewer 3 Report
The topic of the paper is relevant. It addresses the variation in sowing density and nitrogen topdressing in wheat cultivated mode of hole sowing. It was held for two years in China. The results, if properly analyzed and discussed, can generate important information for the positioning of technologies (sowing densities and N topdressing doses) suitable for this cropping system in the region, in addition to contributing to the advancement/consolidation of knowledge on the subject.
In general, the paper presents the necessary items, but several aspects were identified that must be improved in order to reach the necessary technical quality and provide adequate understanding by the reader.
The title could be simplified, as it presents all the treatments and set of variables studied.
The abstract is unbalanced, as it presents little introduction and material and methods and a lot of results and conclusions.
The keywords need to be revised because they present several words that appear in the title. They must include the scientific name of the wheat and other terms not included in the title.
The material and methods need to be supplemented. The levels of treatments involving sowing density need to be transformed into plants.m-2 throughout the paper. Based on the provided arrangement data (hole spacing and number of plants per hole) the densities would be 246, 338, 400 and 492 suitable seeds.m-2. Despite being widely used in practice, the quantification of sowing density in kg.ha-1 is not very accurate for scientific work with plant arrangement, as the same amount in kg.ha-1 can present very large variations in the number of seeds.ha-1 depending on the germination power, weight of a thousand grains, among other factors. Related to this issue, the quantified variables (as grains.m-2) must also be presented on a per m2 basis so that it is possible to relate directly to the grain yield in kg.ha-1. This facilitates interpretation by the reader and reduces the risk of obtaining untrue effects. For example, dry matter per spike must be converted to values ​​per m2.
Some aspects of management and environmental conditions can be improved/justified to facilitate understanding by readers unfamiliar with the production systems used in China. For example, the fact of obtaining grain yields greater than 10,000 kg.ha-1 with a rainfall of just 84.57 mm and 112.68 mm during the cycle. Was irrigation performed during the conduction of the tests? Is there water availability in the soil profile to meet the needs of the crop?
It is necessary to review the statistical analysis and the format for presenting the results, with an impact on the text and on the tables and figures presented. As a rule, when the interaction is significant, the analysis of the interaction is deepened and the simple effect of density or dose of nitrogen is discarded. As it stands, the text presents the analysis of simple effects and, superficially, of interactions. This can confuse the reader in addition to using space in the results with unnecessary information. In addition to this fact, assertions are made based on absolute values, without statistical support. For example, indicating superiority of one treatment over the other and mentioning, in the same sentence, that there was no significant difference. It is considered more appropriate to use regression and not the comparison of means for structured quantitative treatments (density and doses of N). Through regression, it would be possible to establish the density and N dose of maximum technical efficiency and maximum economic efficiency.
Tables and figures need complements to be self-explanatory. Missing units, the name of the culture, meaning of symbols, description of content, separation of numbers and letters, letters with different formats to facilitate interpretation, among others.
The discussions item does not add much to the reader. It needs to be completely overhauled. Initially it presents citations of studies with density and nitrogen in several situations including other cultures. It is possible to approach the subject in more depth, using references on wheat. Later, in this item, the results are presented again. It would be interesting to discuss the results trying to find answers for the effects of the treatments. For example, why did the increase in density increase the dry matter of ears? Why didn't the increase in nitrogen increase the protein content in the grains?
Some adjustments are suggested to qualify the paper:
- Lines 2-4: very specific and long
- Line 10: single sentence. Which method?
- Line 10: abstract is unbalanced: 7 lines of introduction/materials and methods and 17 of results and conclusions. More details of the material and methods are lacking.
- Line 24 and others: speaks of “panicle” when the correct term for wheat is “ears” or “spikes”.
- Line 27: check the value because it calls low density and the values ​​are those of high density (225).
- Line 29: it is not necessary to use two places after the decimal point for grain yield.
- Lines 33 and 34: avoid using terms already used in the title. Include the scientific name of the wheat.
- Lines 72 to 77: more appropriate for a discussion item than an introduction
- Line 103: The doses are “N” or “urea”? In the abstract they imply that they are from N, but in this line they seem to be from urea.
- Line 112: the figure, as it stands, does not clearly show the hole sowing. Check the spacing values ​​between lines and between holes. As it is, it looks like there are 13.5 cm between lines and 25 cm between holes, different from what is described in line 106. I suggest removing some “plants” in the central region of the figure to make the hole sowing evident. As it is, it looks like cultivation in rows and with the central part cut. I consider the wording on the right side of the figure to be unnecessary. It does not add information. Was the “excellent growth environment in each hole?” I didn't find anything in the paper about this, nor comparing with other arrangements.
- Line 113: Was sowing carried out manually or mechanically?
- Line 119: check the proper use of the term “enzymes”
- Line 132: what is the reason for evaluating grain yield in only 1 m2? Small sample size for this type of assessment. Usually overestimates the values.
- Line 133: Was the initial population counted to verify if the plants actually established were close to the sowing density treatments? I suggest using the sowing density treatment unit in “suitable seeds.m-2” and not in “kg.ha-1”.
- Line 139: Make it clear if interaction occurred and when it occurred, explore interactions and not simple effects.
- Line 146: if it only occurred in the 2019-2020 harvest, why do the letters also compare means in the 2018-2019 harvest? I suggest correcting it.
- Line 149: also occurred in filling and maturing.
- Line 153: The authors only refer to interaction in some stages and not in others. What is the reason? Describe all interactions.
- Line 163: What is the explanation for the result obtained for ear dry matter with an increase in dry matter with an increase in sowing density. Usually the result is the opposite.
- Line 168: avoid comparisons between maximum and minimum. They're too drastic.
- Line 170: In Table 1 there is no information if the letters compare means in the column or in the line. Also, averages within each density level or between density levels are compared. Include space between the value and the letters. The unit of the variable was not indicated. The meaning of asterisks was not included in the statistical analysis. Stages need to be adjusted to be in the correct columns. The table needs to be supplemented to be self-explanatory. Check this aspect in the other tables and figures.
- Line 173: Figure 2 refers to the evaluations carried out at which crop stage? I suggest citing the growing season on each chart.
- Line 178: Figure 3 refers to the evaluations carried out at which crop stage? I suggest citing the growing season on each chart. Was the regression significant in each year? The R2 values ​​are very low and in the 2019-2020 crop chart it seems that there was a lot of variability. If the linear regression was not significant, the lines and equations should not be shown as they lead the reader to think they are significant. What is the coefficient of variation for each test?
- Lines 187 to 189: indicates an importance scale (N4>N1>N3>N2), but there was no significant difference between treatments. It induces the reader to misinterpret the results. It is important to be faithful to the statistical results.
- Lines 189 to 191: It points out differences where they did not occur. It is important to be faithful to the statistical results.
- Lines 191 to 192: Table 2 shows non-significant response for N in one year and interaction in another. Adjust reporting of results.
- Lines 195 to 198: The statement is not supported by the values ​​shown.
- Line 203: replace D with N. You cannot point out a difference without the support of statistics.
- Lines 216 to 223: the description is not fully supported by the results in Table 2.
- Lines 219 to 222: The statement cannot be generalized, as it depends on the variable and the growing season. It is important to explore the interactions.
- Lines 260 to 263: I suggest avoiding comparing extremes and valuing more treatments with the same performance, but with less use of inputs and cost. Alignment with the justification presented in the introduction.
- Lines 265 to 269: Figures 4 and 5 lack the culture (wheat).
- Line 273: if no significant difference was observed between means for thousand grain weight, the letters for comparison should not be presented. The yield component number of grains per ear should be presented in the form of m-2 grains. This can correct distortions due to sowing density treatment.
- Lines 278 to 279: I suggest including the description of the meaning of the colors in figures 6 and 7 and not in the text.
- Line 294: include the same description used in Figure 6 (lines 290 to 292).
- Line 299: In the discussion, do you address more the explanation for the results, such as, for example, why the increase in the dose of N did not increase the protein in the grains in an evident way? Did this occur due to the high base fertilization used? By the cultivation system?
- Lines 332 to 334: How can the hole sowing system be valued if it has not been compared with other arrangement systems used in the region? The assertion made is not supported by the study.
- Lines 330 to 356: it is a repetition of the results. It should deepen the explanations about the results obtained using specific literature on the subject for the wheat crop. I suggest reviewing and improving.
- Line 353: The statement “our research helps to ensure food security” is disconnected from the work. It was not the objective of the work. Review the fact of highlighting only some treatments while others, with lower use of inputs/cost, did not differ in performance.
Author Response
Response to comments
Dear Reviewer
We are grateful to the Journal of Agronomy for providing us with the valuable opportunity to revise and modify our manuscript. We would like to thank the reviewers for their careful review and comments, and for your hard work. All the comments have been carefully revised and highlighted in red color for your convenience. We shall look forward to hearing from you and hope that you will consider our work again and give us the opportunity to communicate further with the reviewers as needed.
Response to reviewer comments:
- The title could be simplified, as it presents all the treatments and set of variables studied.
Response: Thanks for pointing out the flaw in the manuscript,We have simplified the title“The effects of different sowing density and nitrogen topdressing on wheat were investigated under the cultivation mode of hole sowing”(Lines 2-4)
- The abstract is unbalanced, as it presents little introduction and material and methods and a lot of results and conclusions.
Response: Thanks for pointing out the flaw in the manuscript,We have modified and supplemented the abstract.“Hole sowing is a new and efficient cultivation method with few studies. This study investigated the effects of different sowing densities and nitrogen topdressing at the jointing stage on dry matter, quality, and yield under wheat hole sowing to provide a theoretical basis for integrating wheat fertilizer and density-supporting technology. In this study, a two-factor split-plot design was used. The sowing density was the main plot, and four levels were set: D1, D2, D3, and D4 (238, 327, 386, and 386 suitable seeds·m−2), the four sowing levels were sown according to 8 grains/hole, 11 grains/hole, 13 grains/hole, and 16 grains/hole, respectively, with a row spacing of 25 cm and a hole spacing of 13.5 cm; the amount of nitrogen fertilizer applied at the jointing stage was the sub-area, with four levels: N1, N2, N3, and N4 (0, 60, 120, and 180 kg·ha−1). After two years of experimental research, the following main conclusions are drawn: the use of high sowing density and nitrogen topdressing amount is helpful to improve the dry matter quality of wheat spikes at the maturing stage; the sowing density had significant or highly significant effects on protein content, starch content, and sedimentation value. The yield in 2018-2019 reached a maximum of 8448.67 kg·ha−1 under D4N4 treatment, and the yield in 2019-2020 reached a maximum of 10136.40 kg·ha−1 under D4N3 treatment. Therefore, the combination of 225 kg·ha−1sowing density and 120-180 kg·ha−1 nitrogen topdressing at the jointing stage can be used in field production, which can help improve wheat production potential. Similarly, understanding the interaction between wheat hole sowing and different sowing densities and nitrogen topdressing amounts provides a practical reference for wheat high-yield cultivation techniques.”(Lines 10-26)
- The keywords need to be revised because they present several words that appear in the title. They must include the scientific name of the wheat and other terms not included in the title.
Response: Thanks for pointing out the flaw in the manuscript,We have simplified the keywords.“Wheat; Triticum aestivum L; hole sowing; cultivation techniques; yield”(Lines 28)
- The material and methods need to be supplemented. The levels of treatments involving sowing density need to be transformed into plants.m-2 throughout the paper.
Response: Thanks for pointing out the flaw in the manuscript,We have modified the density of the full text.“The main area was sowing density, and four sowing density levels were set: D1 (238 suitable seeds·m −2), D2 (327 suitable seeds·m −2), D3 (386 suitable seeds·m −2), D4 (475 suitable seeds·m −2). ”(Lines 94-96)
- Some aspects of management and environmental conditions can be improved/justified to facilitate understanding by readers unfamiliar with the production systems used in China. For example, the fact of obtaining grain yields greater than 10,000 kg.ha-1 with a rainfall of just 84.57 mm and 112.68 mm during the cycle. Was irrigation performed during the conduction of the tests? Is there water availability in the soil profile to meet the needs of the crop?
Response: Thanks for pointing out the flaw in the manuscript,We have modified and supplemented this part.“During the experiment, the wheat was sown for 10 days, mid-November, March and May of the second year, and irrigated according to the actual situation in the field. The two-year processing was consistent and harvested on June 4, 2019, and June 1, 2020, respectively.”(Lines 109-112)
- It is necessary to review the statistical analysis and the format for presenting the results, with an impact on the text and on the tables and figures presented.
Response: Thanks for pointing out the flaw in the manuscript,We have modified and supplemented this part.
- Tables and figures need complements to be self-explanatory. Missing units, the name of the culture, meaning of symbols, description of content, separation of numbers and letters, letters with different formats to facilitate interpretation, among others.
Response: Thanks for pointing out the flaw in the manuscript,We have supplemented the tables and pictures.“Note: D and N respectively represent seeding rate and nitrogen topdressing; D×N means interaction effects of seeding rate and nitrogen topdressin. Different letters in the same column mean significant difference at 0.05. *,**and *** refer to significantIy different of 0.05 , 0.01 and 0.001 level, the same as following.”(Lines 72-74)
“Figure 6. Correlation analysis of different wheat indexes in 2018-2019 (Different colors in the figure represent positive and negative correlation, and color depth represents the correlation size. The bluer the color, the greater the positive correlation coefficient; the red the color, the greater the negative correlation coefficient. X axis and Y axis represent different indexes, r values in the Figure in different colors, *: p ≤ 0.05, **: p ≤ 0.01, the same as following)”(Lines 273-278)
- The discussions item does not add much to the reader. It needs to be completely overhauled. Initially it presents citations of studies with density and nitrogen in several situations including other cultures. It is possible to approach the subject in more depth, using references on wheat. Later, in this item, the results are presented again. It would be interesting to discuss the results trying to find answers for the effects of the treatments. For example, why did the increase in density increase the dry matter of ears? Why didn't the increase in nitrogen increase the protein content in the grains?
Response: Thanks for pointing out the flaw in the manuscript,We have modified and supplemented the discussions“Sowing density is a limiting factor for plants to obtain environmental resources [29]. It is considered to be one of the most influential cultivation methods for grain yield and other agronomic traits. Changes in sowing density are particularly important in wheat crops, which have a direct impact on grain yield and its components [14]. The dynamics of nitrogen and its loss trend create a challenging environment for the effective management of this nutrient in topdressing [30], which is mainly due to various reactions and instability in the soil. The low efficiency of nitrogen is attributed to the volatilization, leaching, and surface runoff of ammonia [20]. Some studies have found the effects of sowing density and nitrogen on crops. For example Kanwal, A et al. [31], by evaluating the effects of different sowing densities and nitrogen doses on oat forage yield, it was found that the interaction of sowing density and nitrogen amount significantly changed the yield and quality attributes of oat green forage. The sowing rate of forage oat crops should be 90 kg·ha −1 and supplemented with 120 kg·ha −1 nitrogen. Higher yield, better quality, and better return.
Our research group has previously proved that hole sowing has an excellent effect on the growth characteristics of wheat by comparing the wheat hole sowing with the traditional sowing method. Wu et al. [36] studied the effects of different sowing methods (drill sowing, wide sowing, and hole sowing) on the yield and quality of wheat. It was found that the hole sowing treatment increased the flag leaf area of wheat, and the nitrogen application increased the dry matter quality of the above-ground part of the hole sowing treatment, and the actual yield of the hole sowing treatment was the highest. However, most of the field experiments on wheat sowing density and nitrogen topdressing in the past were carried out by drilling technology, and the influence of the hole sowing cultivation method was not explored [32-35]. Under the conditions of this experiment, the density had a very significant effect on the number of effective spikes and yield. Increasing the sowing density would reduce the number of grains per spike and thousand grain weight, significantly increase the number of effective spikes per unit area, and expand the number of populations, which could compensate for the lack of individuals. Increasing the amount of topdressing nitrogen had little effect on protein content and settlement value, which may be due to the high nutrient content in the soil before sowing in this experiment, so topdressing had little effect on experiment. The results of our experiment were also slightly different between years. The dry matter of wheat spike, the number of effective spikes per unit area, yield, protein content, and settlement value of each treatment in 2018-2019 were lower than those in 2019-2020. The main reason may be that in 2019-2020, the precipitation and average temperature during the wheat growth period were higher than in 2018-2019, and abundant rainfall and suitable temperature were conducive to crop growth and development. ”(Lines 285-321)
Response to the specific comments of the reviewer :
- Lines 2-4: very specific and long
- Line 10: single sentence. Which method?
- Line 10: abstract is unbalanced: 7 lines of introduction/materials and methods and 17 of results and conclusions. More details of the material and methods are lacking.
- Line 24 and others: speaks of “panicle” when the correct term for wheat is “ears” or “spikes”.
- Line 27: check the value because it calls low density and the values ​​are those of high density (225).
- Line 29: it is not necessary to use two places after the decimal point for grain yield.
Response: Thanks for pointing out the flaw in the manuscript,We simplified the title and modified the abstract.“Abstract: Hole sowing is a new and efficient cultivation method with few studies. This study investigated the effects of different sowing densities and nitrogen topdressing at the jointing stage on dry matter, quality, and yield under wheat hole sowing to provide a theoretical basis for integrating wheat fertilizer and density-supporting technology. In this study, a two-factor split-plot design was used. The sowing density was the main plot, and four levels were set: D1, D2, D3, and D4 (238, 327, 386, and 386 suitable seeds·m−2), the four sowing levels were sown according to 8 grains/hole, 11 grains/hole, 13 grains/hole, and 16 grains/hole, respectively, with a row spacing of 25 cm and a hole spacing of 13.5 cm; the amount of nitrogen fertilizer applied at the jointing stage was the sub-area, with four levels: N1, N2, N3, and N4 (0, 60, 120, and 180 kg·ha−1). After two years of experimental research, the following main conclusions are drawn: the use of high sowing density and nitrogen topdressing amount is helpful to improve the dry matter quality of wheat spikes at the maturing stage; the sowing density had significant or highly significant effects on protein content, starch content, and sedimentation value. The yield in 2018-2019 reached a maximum of 8448.67 kg·ha−1 under D4N4 treatment, and the yield in 2019-2020 reached a maximum of 10136.40 kg·ha−1 under D4N3 treatment. Therefore, the combination of 225 kg·ha−1sowing density and 120-180 kg·ha−1 nitrogen topdressing at the jointing stage can be used in field production, which can help improve wheat production potential. Similarly, understanding the interaction between wheat hole sowing and different sowing densities and nitrogen topdressing amounts provides a practical reference for wheat high-yield cultivation techniques.”(Lines10-26)
- Lines 33 and 34: avoid using terms already used in the title. Include the scientific name of the wheat.
Response: Thanks for pointing out the flaw in the manuscript,We have modified the part.“Keywords: Wheat; Triticum aestivum L; hole sowing; cultivation techniques; yield”(Lines 27)
- Lines 72 to 77: more appropriate for a discussion item than an introduction
Response: Thanks for pointing out the flaw in the manuscript,We add this part to the discussion.
- Line 103: The doses are “N” or “urea”? In the abstract they imply that they are from N, but in this line they seem to be from urea.
Response: Thanks for pointing out the flaw in the manuscript,We have modified the part.“ The nitrogen fertilizer (nitrogen content 46.4 %), and the base fertilizer was wheat special slow-release fertilizer (N: P2O5: K2O mass fraction 24: 15: 5) 750 kg·ha −1, and the base fertilizer was applied once during rotary tillage.”(Lines 97-98)
- Line 112: the figure, as it stands, does not clearly show the hole sowing. Check the spacing values between lines and between holes. As it is, it looks like there are 13.5 cm between lines and 25 cm between holes, different from what is described in line 106. I suggest removing some “plants” in the central region of the figure to make the hole sowing evident. As it is, it looks like cultivation in rows and with the central part cut. I consider the wording on the right side of the figure to be unnecessary. It does not add information. Was the “excellent growth environment in each hole?” I didn't find anything in the paper about this, nor comparing with other arrangements.
Response: Thanks for pointing out the flaw in the manuscript, We deleted the inaccurate picture.
- Line 113: Was sowing carried out manually or mechanically?
Response: Thanks for pointing out the flaw in the manuscript,We have modified the part.“ Sowing carried out manually on October 5, 2018, and October 1, 2019,”(Lines 106)
- Line 119: check the proper use of the term “enzymes”
Response: Thanks for pointing out the flaw in the manuscript,We have modified the part.“ the wheat spikes were baked in the oven at 105 ° C for 30 minutes,”(Lines 117-118)
- Line 132: what is the reason for evaluating grain yield in only 1 m2? Small sample size for this type of assessment. Usually overestimates the values.
Response: Thanks for pointing out the flaw in the manuscript,We have modified the part.“Due to the small area of the plot, the hole sowing has obvious border effect. In order to eliminate the influence of the border effect on the yield, 1m 2 wheat was randomly taken from each plot in the middle and threshed with a thresher, dried in the sun, and weighed with an electronic balance to calculate the grain yield ( kg · ha − 1 ).”(Lines 132-135)
- Line 133: Was the initial population counted to verify if the plants actually established were close to the sowing density treatments? I suggest using the sowing density treatment unit in “suitable seeds.m-2” and not in “kg.ha-1”.
Response: Thanks for pointing out the flaw in the manuscript,We have modified the part.
- Line 139: Make it clear if interaction occurred and when it occurred, explore interactions and not simple effects.
- Line 146: if it only occurred in the 2019-2020 harvest, why do the letters also compare means in the 2018-2019 harvest? I suggest correcting it.
- Line 149: also occurred in filling and maturing.
- Line 153: The authors only refer to interaction in some stages and not in others. What is the reason? Describe all interactions.
- Line 163: What is the explanation for the result obtained for ear dry matter with an increase in dry matter with an increase in sowing density. Usually the result is the opposite.
- Line 168: avoid comparisons between maximum and minimum. They're too drastic.
Response: Thanks for pointing out the flaw in the manuscript,We modified and supplemented this part.“ The effect of nitrogen topdressing amount on the dry matter under different treatments was highly significant at the Filling stage and Maturing stage in 2018-2019 and 2019-2020, and there were also significant differences between the Heading stage and the Flowering stage in 2019-2020.
At the late Filling stage, the assimilates of wheat plants will be transported to the grains in large quantities, and their dry matter will reach the maximum at the Maturity stage, eventually affecting their yield. Therefore, compared with other growth stages, the dry matter of the wheat Maturity stage has a more significant impact. With the increase of sowing density, the overall performance of dry matter in the Maturing stage was N4 > N3 > N2 > N1; with the increase of the amount of nitrogen, the overall performance of dry matter in the Maturing stage was D4 > D3 > D2 > D1. Because the dry matter of wheat spike in the Maturing stage was the most prominent, different sowing densities and nitrogen topdressing amounts were significantly different at this stage. In our study, we further explored the effects of sowing density (Figure 3) on the dry matter of wheat spikes at the Maturing stage in 2018-2020 by linear regression analysis. Through the analysis of the two-year experiment, it can be found that the sowing density has a significant influence on the dry matter of wheat spike, and it is significant.
The results showed that at the Maturity stage, the dry matter weight of wheat spikes treated with D4N4 was higher than that of other treatments, and the dry matter weight of wheat spikes treated with D1N1 was lower than that of other treatments.”(Lines 148-169)
- Line 170: In Table 1 there is no information if the letters compare means in the column or in the line. Also, averages within each density level or between density levels are compared. Include space between the value and the letters. The unit of the variable was not indicated. The meaning of asterisks was not included in the statistical analysis. Stages need to be adjusted to be in the correct columns. The table needs to be supplemented to be self-explanatory. Check this aspect in the other tables and figures.
Response: Thanks for pointing out the flaw in the manuscript,We have modified the part.
- Line 173: Figure 2 refers to the evaluations carried out at which crop stage? I suggest citing the growing season on each chart.
- Line 178: Figure 3 refers to the evaluations carried out at which crop stage? I suggest citing the growing season on each chart. Was the regression significant in each year? The R2 values are very low and in the 2019-2020 crop chart it seems that there was a lot of variability. If the linear regression was not significant, the lines and equations should not be shown as they lead the reader to think they are significant. What is the coefficient of variation for each test?
Response: Thanks for pointing out the flaw in the manuscript,We have modified the part.“Figure 3. Effects of different sowing densities on dry matter of wheat spikes at Maturity stage from 2018 to 2020”(Lines 179-180)
- Lines 187 to 189: indicates an importance scale (N4>N1>N3>N2), but there was no significant difference between treatments. It induces the reader to misinterpret the results. It is important to be faithful to the statistical results.
- Lines 189 to 191: It points out differences where they did not occur. It is important to be faithful to the statistical results.
- Lines 191 to 192: Table 2 shows non-significant response for N in one year and interaction in another. Adjust reporting of results.
- Lines 195 to 198: The statement is not supported by the values ​​shown.
- Line 203: replace D with N. You cannot point out a difference without the support of statistics.
- Lines 216 to 223: the description is not fully supported by the results in Table 2.
- Lines 219 to 222: The statement cannot be generalized, as it depends on the variable and the growing season. It is important to explore the interactions.
Response: Thanks for pointing out the flaw in the manuscript,We modified and supplemented this part.“By analyzing the effects of different sowing densities and nitrogen topdressing on grain quality (Table 2), from 2018 to 2019, in terms of protein content, D2N4 treatment was the largest, 14.65 %, and D4N3 treatment was the smallest, 13.46 %; with the increase of sowing density, the protein content increased first and then decreased. The protein content was the highest at the D2 sowing density level, which was 14.42 %, and the D2 level was significantly higher than the D3 level. The starch content increased with increased sowing density and nitrogen topdressing amount. Under the conditions of different sowing densities, the settlement value of grains increased first and then decreased with the increase in sowing density. Compared with D1, D3, and D4, D2 increased by 3.50 %, 8.93 %, and 13.22 % respectively. ”(Lines 182-191)
“The results showed that sowing density had significant effects on protein content, starch content, and settlement value but did not significantly affect stabilization time from 2018 to 2020. The amount of nitrogen topdressing had a significant effect on stabilization time. The interaction between sowing density and nitrogen topdressing significantly affected protein content, stabilization time, and settlement value in 2018-2019. In 2019-2020, it significantly impacted stabilization time, starch content, and settlement value. ”(Lines 204-211)
- Lines 260 to 263: I suggest avoiding comparing extremes and valuing more treatments with the same performance, but with less use of inputs and cost. Alignment with the justification presented in the introduction.
Response: Thanks for pointing out the flaw in the manuscript,We modified and supplemented this part.“ Compared with D1, D2, and D3, D4 increased by 17.70 %, 14.53 %, and 10.51 % respectively. There was no significant difference among D1, D2, and D3 levels. The yield increased first and then decreased with the increase of nitrogen application. D4N3 treatment reached the maximum value of 10136.40 kg · ha −1. ”(Lines 244-248)
- Lines 265 to 269: Figures 4 and 5 lack the culture (wheat).
Response: Thanks for pointing out the flaw in the manuscript,We modified and supplemented this part.“ Figure 4. Effects of different sowing density and nitrogen topdressing on wheat yield in 2018 to 2019”(Lines 250-251)
“ Figure 5. Effects of different sowing density and nitrogen topdressing on wheat yield in 2019 to 2020”(Lines 254-255)
- Line 273: if no significant difference was observed between means for thousand grain weight, the letters for comparison should not be presented. The yield component number of grains per ear should be presented in the form of m-2 grains. This can correct distortions due to sowing density treatment.
Response: Thank you very much for your suggestion. This part is presented because the papers published by us and most institutions have always been in this form and have been recognized.
- Lines 278 to 279: I suggest including the description of the meaning of the colors in figures 6 and 7 and not in the text.
- Line 294: include the same description used in Figure 6 (lines 290 to 292).
Response: Thanks for pointing out the flaw in the manuscript,We modified and supplemented this part.“Figure 6. Correlation analysis of different wheat indexes in 2018-2019 (Different colors in the figure represent positive and negative correlation, and color depth represents the correlation size. The bluer the color, the greater the positive correlation coefficient; the red the color, the greater the negative correlation coefficient. X axis and Y axis represent different indexes, r values in the Figure in different colors, *: p ≤ 0.05, **: p ≤ 0.01, the same as following)”(Lines 273-278)
- Line 299: In the discussion, do you address more the explanation for the results, such as, for example, why the increase in the dose of N did not increase the protein in the grains in an evident way? Did this occur due to the high base fertilization used? By the cultivation system?
- Lines 332 to 334: How can the hole sowing system be valued if it has not been compared with other arrangement systems used in the region? The assertion made is not supported by the study.
Response: Thanks for pointing out the flaw in the manuscript,We modified and supplemented this part.“Our research group has previously proved that hole sowing has an excellent effect on the growth characteristics of wheat by comparing the wheat hole sowing with the traditional sowing method. Wu et al. [36] studied the effects of different sowing methods (drill sowing, wide sowing, and hole sowing) on the yield and quality of wheat. It was found that the hole sowing treatment increased the flag leaf area of wheat, and the nitrogen application increased the dry matter quality of the above-ground part of the hole sowing treatment, and the actual yield of the hole sowing treatment was the highest. However, most of the field experiments on wheat sowing density and nitrogen topdressing in the past were carried out by drilling technology, and the influence of the hole sowing cultivation method was not explored [32-35]. Under the conditions of this experiment, the density had a very significant effect on the number of effective spikes and yield. Increasing the sowing density would reduce the number of grains per spike and thousand grain weight, significantly increase the number of effective spikes per unit area, and expand the number of populations, which could compensate for the lack of individuals. Increasing the amount of topdressing nitrogen had little effect on protein content and settlement value, which may be due to the high nutrient content in the soil before sowing in this experiment, so topdressing had little effect on experiment. The results of our experiment were also slightly different between years. The dry matter of wheat spike, the number of effective spikes per unit area, yield, protein content, and settlement value of each treatment in 2018-2019 were lower than those in 2019-2020. The main reason may be that in 2019-2020, the precipitation and average temperature during the wheat growth period were higher than in 2018-2019, and abundant rainfall and suitable temperature were conducive to crop growth and development.”(Lines 299-321)
- Lines 330 to 356: it is a repetition of the results. It should deepen the explanations about the results obtained using specific literature on the subject for the wheat crop. I suggest reviewing and improving.
- Line 353: The statement “our research helps to ensure food security” is disconnected from the work. It was not the objective of the work. Review the fact of highlighting only some treatments while others, with lower use of inputs/cost, did not differ in performance.
Response: Thanks for pointing out the flaw in the manuscript,We modified and supplemented this part.“After two years of research on the use of different sowing densities and nitrogen topdressing amounts of wheat under hole sowing conditions, we found that field production can use a nitrogen density combination of 475 suitable seeds·m −2 sowing density and 120 - 180 kg·ha -1 nitrogen topdressing at the jointing stage, which can fully tap the production potential of wheat. The experimental results fill the gap in wheat research under the cultivation method of hole sowing and provide valuable references and help for future researchers. In addition, there are still some limitations and deficiencies in this experimental study. The experiment is only a two-year research period, and due to the significant difference in climatic conditions between the two years, although the overall trend is consistent, the regularity and universality of individual index changes are not strong. It is necessary to further carry out long-term positioning experiments to more accurately grasp and lay a theoretical basis and technical support for fully tapping wheat's high-quality and high-yield potential under hole sowing conditions.”(Lines 323-337)
In conclusion, we have tried our best to improve the manuscript and have made comprehensive changes in this paper. After reading the latest version of our manuscript, we hope you will have a better understanding of this manuscript. We earnestly appreciate the reviewers’ work and we hope that the changes and corrections will meet with your approval. Once again, thank you very much for your comments and suggestions.
Yours sincerely,
The authors
Reviewer 4 Report
Dear Authors,
thank you for the manuscript draft entitled "The effects of different sowing density and nitrogen topdressing on dry matter, quality, and yield of wheat were investigated under the cultivation mode of hole sowing".
My comments, suggestions, and views on this article are as follows:
TITLE
Comment(s): The title is comprised of the study process in full, its' purpose, focus and area of interest.
Suggestion(s): -
GRAPHICAL ABSTRACT (Optional)
Suggestion(s): the authors may consider submitting graphical abstract. Suggested contents for the graphical abstract: i) study overview/ Purpose of the study, and/or ii) Summary of Key findings for all objectives.
ABSTRACT - adequate
Suggestion(s): -
KEYWORDS
Comment(s): Good keywords in terms of number of words/ terms provided, brevity and do reflect the study, but try not to repeat words from the title.
Suggestion(s): -
INTRODUCTION
Comment(s): This section has a good structure, cohesion and clarity. It includes the background of the study, issues, knowledge gap and practice deficiency, study goals, and research significance. This section also provides a relevant literature, an introduction to the methods used in this study.
Suggestion(s): -
METHODS/ MATERIALS - adequate
Suggestion(s): -
RESULTS, DISCUSSION - quite fine
Suggestion(s): -
CONCLUSION - quite nice but
Suggestion(s): - in my opinion there should be more numerical results
OVERALL: This is a good paper in terms of i) issues highlighted, ii) previous studies reviewed, iii) methods used, iv) findings and discussion, v) contributions to practice and body of knowledge, as well as.
Thanks and best regards.
Author Response
Response to comments
Dear Reviewer
We are grateful to the Journal of Agronomy for providing us with the valuable opportunity to revise and modify our manuscript. We would like to thank the reviewers for their careful review and comments, and for your hard work. All the comments have been carefully revised and highlighted in red color for your convenience. We shall look forward to hearing from you and hope that you will consider our work again and give us the opportunity to communicate further with the reviewers as needed.
In conclusion, we have tried our best to improve the manuscript and have made comprehensive changes in this paper. After reading the latest version of our manuscript, we hope you will have a better understanding of this manuscript. We earnestly appreciate the reviewers’ work and we hope that the changes and corrections will meet with your approval. Once again, thank you very much for your comments and suggestions.
Yours sincerely,
The authors